# Linking Cognitive Reappraisal and Expressive Suppression to Mindfulness: A Three-Level Meta-Analysis

**DOI:** 10.3390/ijerph20021241

**Published:** 2023-01-10

**Authors:** Senlin Zhou, Yunpeng Wu, Xizheng Xu

**Affiliations:** 1Investigation Department, Hunan Police Academy, Changsha 410138, China; 2School of Teacher Education, Dezhou University, Dezhou 253023, China

**Keywords:** cognitive reappraisal, expressive suppression, mindfulness, three-level meta-analysis

## Abstract

Cognitive reappraisal and expressive suppression have long been considered the two most important emotion regulation strategies. Increasingly, studies have found that mindfulness has a relationship with them. However, the magnitude and direction of the relationship between them have yet to reach a consistent conclusion. To this end, a three-level meta-analysis was used to explore the relationship between mindfulness, cognitive reappraisal, and expressive suppression. Thirty-six studies were included in the meta-analysis through a literature search, including 36 samples with 83 effect sizes and a total of 12,026 subjects. The three-level random effects model showed that mindfulness was positively correlated with cognitive reappraisal to a low to moderate degree but was not correlated with expressive suppression. The moderating effect analysis showed that the relationship between mindfulness and cognitive reappraisal was slightly closer in males than in females. The study found a relatively stable positive relationship between mindfulness and cognitive reappraisal, an adaptive emotion regulation mode, while the relationship with expressive suppression was insignificant. The finding supported the affect regulation training model and also shed light on potential areas for future studies.

## 1. Introduction

Emotional regulation refers to the various influencing ways individuals adopt to modulate the generation, process, and result of emotional events, consciously or unconsciously [1], which is a crucial way to adapt to context. Successful emotional regulation can improve happiness, task performance, interpersonal relationships, and other positive psychological and behavioral results [2,3,4], while failed emotional regulation is significantly positively correlated with psychological and behavioral problems such as anxiety, addiction, and depression [5,6,7,8]. Individuals will use various strategies in emotion regulation, and the appropriate choice of these strategies will directly affect the results of emotion regulation. Among these strategies, cognitive reappraisal and expressive suppression are the two most critical moderating strategies [9,10]. Therefore, exploring the influencing factors of these two strategies to promote individual physical and mental health has always been an important concern of clinical psychologists and field researchers [11,12].

In recent decades, mindfulness attracted increasing attention among the factors affecting emotion regulation strategies [13,14]. However, two inconsistent conclusions exist in studying the relationship between mindfulness and emotion regulation strategy. The first is the definition of the conceptual relationship between them. Some researchers have pointed out that mindfulness is a kind of emotional regulation strategy [15]. Accordingly, emotional regulation includes mindfulness. Other researchers insist mindfulness has fundamentally different characteristics from emotion regulation strategies [16]. For instance, Chambers argues that mindfulness differs from cognitive reappraisal and all adaptive emotion-regulation strategies. Emotional regulation must somehow affect the creation, process, or outcome of emotion to change the process or outcome of the emotion [16], while mindfulness emphasizes that cognition and emotion are only mental events and do not need to be acted upon. The second is the controversy over the magnitude and direction of the relationship between mindfulness and other emotion regulation strategies [17]. Most studies have shown that mindfulness can reduce some difficulties in emotional regulation, thus promoting successful emotional regulation [18,19]. Other studies suggested that mindfulness is positively correlated with some adaptive emotion regulation strategies, such as cognitive reappraisal, while it is negatively correlated with other non-adaptive emotion regulation strategies, such as expressive suppression [20,21].

Although researchers have conducted a meta-analysis and systematic review of the relationship between mindfulness and other emotion regulation strategies, some things could be improved [17,22]. First, mindfulness in the meta-analysis mainly includes two dimensions of mindfulness: conscious action and non-judgment. In the definition of mindfulness, many researchers include other dimensions besides the above two. For example, the five-factor mindfulness scale measures include observing, describing, acting with awareness, do not judge, and do not react dimensions [23]. Although some dimensions of mindfulness can be used as emotion regulation strategies, there are more extensive contents in addition to these dimensions. The relationship between mindfulness and emotion regulation strategies still needs to be explored. Second, the meta-analysis only included English literature, and its conclusions were more premised on a Western cultural background. However, mindfulness originates from meditation in Eastern Buddhism, and its connotation, structure, and other characteristics may differ from Western cultural values [24]. Studies with more Chinese subjects representing typical oriental cultural characteristics can compare the Eastern and Western typical groups and obtain more accurate conclusions. Therefore, the meta-analysis method is adopted to conduct a quantitative analysis of the relationship between the two. The detailed integration of the relationship between the respective dimensions helps clarify the differences in the connection between various emotion regulation strategies and the specific dimensions of mindfulness and deepens the understanding of the relationship between the two. It is also helpful to improve the theories related to emotion regulation and mindfulness and can provide a reference for improving emotion regulation through mindfulness training.

### 1.1. Concept and Measurement of Cognitive Reappraisal and Expressive Suppression

In emotion regulation, individuals can change their emotions in physiological activities, subjective experience, and facial behavior through specific strategies [25]. Some researchers have pointed out that any psychological and behavioral activities aimed at affecting emotions belong to the category of emotion regulation strategies [17]. Therefore, emotional regulation strategies include various psychological and behavioral actions related to changing mood, such as watching TV, eating, diversifying attention, ruminating, and so on. Among these strategies, the two most common and valuable strategies are cognitive reappraisal and expressive suppression [9], which are also the strategies that researchers pay the most attention to. Since the literature on the relationship between other emotion regulation strategies and mindfulness is still in the initial stage, the meta-analysis needs to reach a stable conclusion. Therefore, this meta-analysis mainly explores the relationship between cognitive reappraisal, expressive suppression strategies, and mindfulness.

According to the process model of emotion regulation proposed by Gross [26,27], cognitive reappraisal refers to the cognitive change of an individual’s understanding of an emotion-inducing situation or event, thus changing the emotional experience, which occurs in the early stage of the emotional generation process. Expressive suppression refers to the suppression of the current or upcoming emotional explicit behavior, which occurs in the late stage of the emotional generation process. It is generally believed that cognitive reappraisal strategies can produce better results than expressive suppression, and it is also the primary treatment strategy of cognitive therapy [28]. For example, cognitive reappraisal can better reduce the emotional experience. Although expressive suppression can reduce external emotional behavior, it does not mitigate emotional experience [29]. Cognitive reappraisal can reduce the physiological response and the activation of the sympathetic nervous system, while expressive suppression may enhance the activation of the physiological response and sympathetic nervous system [30]. The effect between the two strategies is not absolute but depends on the context. Expressive suppression may have negative consequences in some cases, but this is not always the case. Individuals live in a diversified interpersonal context. It is a common strategy to adapt to the external environment by suppressing the expression of emotions, which has crucial positive significance for individuals to adjust to the external environment and promote interpersonal harmony.

Researchers have developed various methods to measure cognitive reappraisal and expressive suppression. These measures can mainly be divided into three categories: One is to measure the trait emotional regulation through the emotional regulation scale [31]. The second is to measure the state emotional regulation through the emotion regulation task set up in the laboratory [32]. Third, the implicit association test was used to measure the implicit emotional regulation of the subjects [33]. The first measure can measure the individual’s relatively stable tendency to use cognitive reappraisal, expressive suppression, or other strategies. The emotion regulation questionnaire (ERQ) [34] is widely used. The scale consists of 10 items, including six for the cognitive reappraisal dimension and four for the expressive suppression dimension. The scoring method was a seven-point Likert scale, translated into multiple language versions, and has good cross-regional, cross-ethnic, and cross-cultural applicability [35]. The second method mainly explores the strategies and influencing factors of emotion regulation induced by the situation. The third method measures implicit emotion regulation in individuals. As this study focuses on mindfulness and emotion regulation strategies as stable traits, the studies included in the meta-analysis mainly include individuals’ relatively stable emotional regulation strategy tendencies, which were mostly collected from the first measurement method.

### 1.2. Concept and Measurement of Mindfulness

Originating from Buddhism, mindfulness initially refers to stabilizing the mind, being clearly and intently aware of what is happening in the body, and neither forgetting nor letting it disappear [36]. With the deepening of the research on mindfulness, the study of mindfulness gradually transcends the fields of Buddhism and psychology [37], and its concept is also evolving. Mindfulness is a concept with multiple meanings. In addition to thinking of it as a physical practice, known as meditation, other researchers hold mindfulness as a state of mind, a mental process, or a quality. Kabat-Zinn defined it as the awareness that arises through attention on purpose, in the present moment, and non-judgmentally [38]. Therefore, in primary psychology, mindfulness is not a single term. Some researchers understand mindfulness from the trait, state, and intervention aspects. Trait mindfulness is the individual difference between individuals in their ability and tendency to experience the state of mindfulness, which is relatively stable and not easy to change [39]. State mindfulness is a state of consciousness in the body that changes with the situation and time [40]. Mindfulness intervention is a training method to help people achieve the state of mindfulness [41]. Long-term training can improve the level of trait mindfulness. In theoretical research, most scholars regard mindfulness as a general trait, state, and intervention term. However, in the empirical research, they are further subdivided due to the different research methods of the three. This study focuses on trait mindfulness, which is mainly measured by scales. The most widely used scales to measure mindfulness are the Mindfulness Attention Awareness Scale (MAAS) [42] and the Five Facet Mindfulness Questionnaire (FFMQ) [43]. MAAS emphasizes the importance of awareness in mindfulness. The scale has 15 items and adopts a six-level scoring method, which belongs to the single-factor structural scale. FFMQ consists of 39 items and adopts a five-level scoring method, including five factors: observation, to describe, to act with awareness, not to judge, and no response.

### 1.3. The Relationship between Mindfulness and Cognitive Reappraisal and Expressive Suppression

A cognitive-oriented view of mindfulness is based on the framework of information processing, which regards mindfulness as a cognitive process, holding that individuals process internal and external experiences with a non-judgmental attitude, thus transforming the way of thinking, which can promote the occurrence of cognitive reappraisal [44]. Many empirical studies have also confirmed the positive relationship between them. For example, Hanley and Garland used 812 normal American adults, contemplative practitioners, and other patients as subjects to detect the relationship between dispositional mindfulness and reappraisal. The results indicate that dispositional mindfulness positively relates to self-reported positive reappraisal crossing all five samples [45]. Naragon–Gainey and colleagues conducted a meta-analysis on ten emotion regulation strategies based on a 667 effect size. The result found that the pooled Pearson correlation coefficient *r* of mindfulness and cognitive reappraisal is 0.36 [17]. Based on the literature mentioned above, it is speculated that mindfulness is positively correlated with cognitive reappraisal strategies.

Findings from neuron physiological empirical studies reveal the relationship between mindfulness and expressive suppression. For example, the volume of gray matter in the right orbitofrontal cortex and right hippocampus of long-term mindful meditation practitioners increased significantly, both of which are involved in emotional regulation and response control [46], indicating that mindfulness training can improve response control ability, but whether this control ability can facilitate expressive suppression has not been experimentally explored. On the contrary, many questionnaire studies implicated mindfulness’s negative relationship with expressive suppression [17]. From the above controversial evidence, it cannot indicate the “real” relationship between mindfulness and expressive suppression. So, we only detect their relationship for an exploratory aim and have no hypothesis.

### 1.4. Possible Moderating Variable

**Age.** Different ages may influence the correlation between mindfulness and emotion regulation. This age difference may be caused by psycho-social development factors and life goal selection factors [47]. On the one hand, with the continuous maturation of body and mind and the enrichment of interpersonal experience, individuals are more inclined to adopt conscious strategies to regulate emotions. Therefore, mindfulness can be used consciously to regulate emotions. On the other hand, according to socioemotional selectivity theory (SST), people’s motivation and target shifted from knowledge acquisition to emotional satisfaction acquisition [48]. This increased emphasis on emotion will also encourage individuals to strengthen the relationship between mindfulness and the ability to regulate emotions. Because of these age differences in individuals, the study speculates that different age groups may influence the relationship between mindfulness and emotional regulation.

**Gender.** Gender may influence the relationship between mindfulness and emotional regulation due to different social expectations of male and female gender roles. We generally believe that men are reserved in expressing emotions, while women can have more emotional behaviors. Because of this gender difference in emotional expression, they tend to use different emotional regulation strategies. For example, men use more expressive suppression than women to regulate negative emotions [49]. However, it remains to be explored whether the above differences in emotional regulation strategies extend to the relationship between mindfulness and cognitive reappraisal and expressive suppression.

**Measurement tools.** Different measurement tools may affect the relationship between mindfulness and emotion regulation strategies. In mindfulness and emotion regulation research, different dimension structures correspond to different measurement tools. MAAS believes that focusing attention and being aware of the present are the core of mindfulness. The scale regards mindfulness as a single structure, only measuring the degree of maintaining awareness and concentration, but ignoring other aspects of mindfulness, such as acceptance, non-judgment, and non-reaction. The FFMQ considers mindfulness a skill to measure an individual’s propensity to use it in their daily lives. The same issues may also exist in the research on emotion regulation. The different dimensions measured using different tools, such as the above questionnaires, may influence the magnitude and direction of the effect size of the relationship between mindfulness and the above two emotion regulation strategies. According to the above states, both the mindfulness and emotional regulation measurement tools rate the relationship between mindfulness and cognitive reappraisal or expressive suppression.

**Culture.** Different cultural groups can also influence the relationship between mindfulness and emotion regulation. Firstly, more and more studies have shown that expressive suppression has a culture-specific effect on the regulation of negative emotions [50]. In the context of Western individualistic culture, emotional expression is advocated. Compared with Eastern culture, the suppression of emotional expression will bring more negative results. Meanwhile, in the context of Eastern culture, expressive suppression is more conducive to improving social functions [51]. Secondly, mindfulness originated from the meditation of Eastern Buddhism, which may be more in line with the thinking mode of Oriental people. Moreover, studies have found that the effect of mindfulness training on Oriental people is better than that of Westerners [52]. Therefore, mindfulness may be more closely associated with cognitive reappraisal and expressive suppression in Eastern cultures than in Western cultures. In conclusion, due to the differences in the effects of mindfulness training and the use of emotion regulation between Eastern and Western cultures, it is speculated that subjects with different cultural backgrounds may affect the relationship between mindfulness and emotion regulation strategies.

### 1.5. Current Research

Given the differences in previous empirical studies, and as a further exploration of the conclusion of a recent meta-analysis, our study raises two questions: First, what is the magnitude and direction of the relationship between each dimension of mindfulness and cognitive reappraisal and expressive suppression? Second, what factors influence the relationship between the dimensions of mindfulness and cognitive reappraisal and expressive suppression? To answer these two questions, the current study uses the three-level meta-analysis method to integrate previous empirical studies quantitatively and tests the mindfulness dimension, gender, age, culture, and other factors as possible moderating variables affecting the relationship, to reveal the “true” relationship between mindfulness and cognitive reappraisal and expressive suppression. In addition, according to previous studies, in the process of testing moderating variables, due to the lack of sufficient basis to assume the influence of other variables except age and culture on mindfulness, cognitive reappraisal, and expressive suppression, this study only conducted an exploratory test of their moderating effect without making specific assumptions.

Based on the above review, the following hypotheses are proposed:

**Hypothesis** **1.**
*Mindfulness is significantly positively correlated with cognitive reappraisal.*


**Hypothesis** **2.**
*The relationship between mindfulness and cognitive reappraisal and expressive suppression is moderated by age, gender, culture, and mindfulness and emotion regulation measurement tools.*


## 2. Materials and Methods

### 2.1. Literature Search and Screening

Following the Preferred Reporting Items for Systematic Reviews and Meta-Analyses (PRISMA) approach, Chinese papers were searched in the CNKI database, WiP Chinese Science and Technology Journal, and Wanfang Data Retrieval system. We searched for foreign papers in the Web of Science, PubMed, and the Elsevier Science Direct database for keywords. The first and the third authors extracted the high-frequency keywords in the databases, and other keywords were evaluated independently by those two authors. All discrepancies were resolved via discussion. The search string included the following elements: “mindfulness, cognitive reappraisal, and expressive suppression”. Synonymous terms such as “mindful*, meditate*, emotion regulation, emotion adjust, express*, suppress*” were combined with the Boolean “OR”. These three concepts were then combined with the Boolean “AND”. The search time range is unlimited. The documents were imported in Excel and screened according to the following criteria: (1) The papers must be empirical papers that conduct quantitative analysis on the relationship between mindfulness and emotion regulation, excluding theoretical, review, and interview papers. (2) The Pearson correlation coefficient r was used as the effect size in these studies. So, the correlation coefficient r between mindfulness and cognitive reappraisal or expressive suppression and its sample size or statistics such as F, t, or Chi-square that could be converted into r must be reported in these studies. (3) The subjects were the general group, and the special clinical groups, such as patients with depression and mental illness, were not included. (4) Research data shall not be reused. If the dissertation is published in an academic journal, the published journal paper shall prevail. (5) Each independent sample is coded once. If a document contains multiple independent samples, it will be coded separately. If the samples are not independent, such as a sample with more than one effect size, then multiple codes are made. Feature coding was carried out for valid literature samples, including literature title, journal, author, publication year, and other information, reliability and validity of the variable measurement, sample size, and other statistics. The first and the third authors independently conducted the abstract and full-text screening, and any disagreements on the criterion were resolved by consensus. A total of 2936 articles were searched; 819 duplicated articles were excluded according to the above criteria, and 1839 studies did not meet the inclusion criteria (irrelevant subject matter, review category, and subjects’ characteristics did not meet the inclusion criteria). The full text of five articles was not found, and 62 articles have no complete data. Then, 207 articles were excluded through full-text reading (research variables were irrelevant to the subject matter), and 36 articles with 83 effect sizes were finally determined to be included. A total of 12,026 subjects were included. The literature inclusion process is shown in Figure 1.

### 2.2. Coding Process

**Mindfulness coding.** According to the currently widely recognized method of dividing the dimensions of mindfulness, mindfulness in the study is coded into six categories: single-dimensional mindfulness, and five dimensions of mindfulness, namely, observation, description, acting with awareness, non-judgment, and non-reaction.

**Cognitive reappraisal and expressive suppression coding.** Due to the different concerns of different emotional regulation measurement tools, there are many dimensions of measurement content, but this study focuses on cognitive reappraisal and expressive suppression dimensions. Therefore, only emotional regulation strategies involving cognitive reappraisal or expressive suppression were coded in this study. Age was coded as a continuous variable, while cultural background East and West were coded as category variables together with measurement method, gender, and publication status. Countries that fail to be categorized clearly as East and West are not counted in the code. The first and the third authors double-coded the data and used statistical magnitude Kappa and ICC to estimate the degree of consistency. The final coding is shown in Appendix A.

### 2.3. Quality Rating

Given that the majority of studies included in our meta-analysis are cross-sectional questionnaire studies, the methodological quality of eligible articles was evaluated with the critical appraisal tool offered by Cortés-García et al. [53]. The appraisal tool has nine items (e.g., clarity of study aims, sample representativeness, and appropriate design). If a study meets one item, it will receive a score of 1. Then, studies with a score lower than 3 were divided into the weak category, studies with a score between 4 and 6 were divided into the middle category, and studies with a score between 7 and 9 were divided into the strong category. Disagreements on the judgment were resolved by consensus. Finally, there were 5 studies in the weak category group, 24 in the middle category group, and 7 in the strong category group. We take the quality score as a potential moderator to detect if it can influence the results of the pooled effect sizes.

### 2.4. Analysis Process

Since many of the studies included in the meta-analysis have multiple effect sizes from the same sample, this violates the assumption of effect size independence of meta-analysis. Because the correlation between effect sizes may lead to standard estimation bias, false inferences can be generated [54]. The three-level meta-analysis can solve the problem related to effect size. This method can calculate variances at the individual level (level 1), intra-study level (level 2), and inter-study level (level 3). If other conditions are equal, the research that provides more effect size and the research that provides less effect size should have the same weight. At the same time, it can include all effect sizes, generate higher statistical power, and provide more accurate estimates than traditional methods [55].

Data analysis was conducted using R studio (R Core Team, 2016) based on R language version 4.2.2, metafor function package [56,57], and Psychmeta function package [58]. The correlation coefficient *r* was used to integrate the relationship between variables. First, to reduce the estimation deviation caused by the internal consistency reliability of the scale, *r* was corrected. The correction formula was [58]:(1)$$\rho =r\mathrm{xy}/\sqrt{r\mathrm{xx}*r\mathrm{yy}}\;(r\mathrm{xy}=r;\;r\mathrm{xx}=\mathrm{Cronbach}’s\;a\mathrm{of}\mathrm{variable}x;r\mathrm{yy}=\mathrm{Cronbach}’s\;a\mathrm{of}\mathrm{variable}y)$$

Furthermore, due to the *r* magnitude not being normally distributed, it was translated to Fisher’s *z* before formal analysis and then back-translated, pooling Fisher’s *z* to *r*. All the meta-analysis results showed the corrected relationship values. Second, we establish the three-level and two-level random effect models (including the level 1 and 2 models and the level 1 and 3 models) and compare the three-level model with the two-level model to determine the optimal model. Thirdly, according to the classical meta-analysis method, publication bias was tested by funnel plot, and the symmetry of the funnel plot was tested by Eggers regression [59]. If bias exists, the trim-fill method of the classical meta-analysis will be used to correct the studies with bias [60]. Fourth, the combined effect size is calculated for the meta-analysis of trim-fill correction, and the uncorrected and corrected meta-analysis results are reported. Finally, the heterogeneity of the model is analyzed. If the heterogeneity test index *Q* is significant (*p* < 0.1) or *I*^2^ > 75% [61,62], then meta-regression analysis was used to examine the moderating effects of various co-variables.

## 3. Results

### 3.1. Publication Bias Test

First, the Kappa and ICC value are from 0.92 to 0.98, indicating high coding reliability. Second, sensitive analysis using Cook’s distance index found no outlier effect size. Third, publication bias was visually observed through the symmetry of the funnel plot (Figure 2) effect size. Fourth, the Eggers test is carried out by adding sample variance as a co-variable to the random effects model to solve the non-independent problem of multiple effect sizes. Eggers test was performed on the total sample and each subgroup, and it was found that there was no significant publication bias in the meta-analysis of mindfulness and cognitive reappraisal (*z* = 0.12, *p* = 0.90). While there was a publication bias between mindfulness and expressive suppression meta-analysis (*z* = 2.28, *p* = 0.02), the trim-fill method was used in the subsequent analysis to obtain the adjusted pooled effect size.

Furthermore, to verify the robustness of the pooled effect size gained from the trim-fill method and to offer a more precise effect size in the publication bias circumstance, we add the pooled effect size obtained from the robust Bayesian meta-analysis method conducted by JASP software [63,64]. We found the absence of evidence for the presence of the effect *BF*_10_ = 0.28 and considerable evidence for publication bias, *BF*_pb_ = 15.79. The same result from the trim-fill method, the pooled effect size obtained by the robust Bayesian meta-analysis method, is insignificant (see Table 1).

### 3.2. Model Comparison

Given that the study samples included in our meta-analysis are from different cultures, ages, and backgrounds, other potential moderators may induce heterogeneity, so it is reasonable to conduct a random effects model. The three-level random effects model was significantly better for the relationship between mindfulness and expressive suppression than the two-level model. Furthermore, according to the values of *Q* and *I*^2^ (see Table 1), it is reasonable to adopt the random effects model. It is also revealed that there may be moderating variables leading to heterogeneity. For mindfulness and cognitive reappraisal, there was no difference according to the fit index between the three-level random effects model and the level 1 and 2 random effects model. This indicates that the correlation between multiple effect sizes obtained from the same sample is not large enough to significantly affect the accuracy of the combined effect size (see Table 2). Therefore, in the meta-analysis of mindfulness and cognitive reappraisal, both three-level meta-analysis and traditional meta-analysis results are presented (see Table 1). According to the values of *Q* (*p* < 0.01) and *I*^2^ (>75%), heterogeneity exists and the random effects model is reasonable. It is further revealed that there may be moderating variables leading to heterogeneity.

### 3.3. Pooled Effects

The pooled effect size showed that mindfulness was significantly correlated with cognitive reappraisal, *r* = 0.23, 95%CI = [0.14, 0.31], *p* < 0.01. Evidence from both the trim-fill method and the robust Bayesian meta-analysis method certified that there was no significant correlation between mindfulness and expressive suppression. Hypothesis 1 is supported.

### 3.4. Analysis of Moderating Effects

To ensure meta-regression stability, variables with less than four effect sizes were not included in the regression analysis [65]. We conduct meta-regressions analysis with gender, age, mindfulness dimension, culture, and measurement tool (category variable virtual coding, continuous variable centralization) as independent variables and effect size as the dependent variable, respectively. The moderating effect of gender on the relationship between mindfulness and cognitive reappraisal was marginally significant (*b* = −0.02, *p* = 0.09). The moderating effect of mindfulness measurement tools was significant in the relationship between mindfulness and expressive suppression (*b* = −0.35, *p* < 0.01). The relationship between mindfulness and expressive suppression measured by the FFMQ was significantly higher than that measured by MAAS. The moderating effects of other variables were not significant. The main results are shown in Table 3. Hypothesis 2 was partially supported.

## 4. Discussion

The results of our study showed that mindfulness is related to cognitive reappraisal in a relatively stable form, even though the magnitude of the relationship may be slightly verified for different genders. Specifically, the relationship is closer for men than for women. These results are similar to previous studies and meta-analyses [17,44]. We also found that mindfulness has no relationship with expressive suppression, which is different from some previous studies. Generally, the results extended theories of emotion regulation, such as the affect regulation training (ART) model [66], which argued that some emotion regulation strategies have similar effects and results are closer than other strategies. Accordingly, mindfulness and cognitive reappraisal can be categorized as adaptive emotion regulation strategies, while expressive suppression belongs to an ill-adaptive strategy. The results also confirm the cognitive-oriented view of mindfulness which emphasizes the cognitive function of mindfulness [67].

On the one hand, our result of the insignificant relationship between mindfulness and expressive suppression is similar to some previous studies. For instance, a self-report measure study with 187 Chinese emerging adults found no significant relationship between mindfulness and expressive suppression [68]. On the other hand, other studies found a negative relationship between them [17]. The main reasons for this inconsistency may be as follows. First, publication bias caused by incomplete literature inclusion affected the accuracy of the results. Second, expressive suppression may be influenced by more environmental factors than cognitive reappraisal [69,70], and these environmental factors may act as moderators which have not been detected. Third, the studies included in the current meta-analysis only included trait mindfulness less influenced by the environment. Finally, the limited studies included in our meta-analysis may cause low statistical power and type II error. So, the result of no correlation between mindfulness and expressive suppression should be cautiously explained. The inconsistent conclusions also call for more research in this field.

### 4.1. The Moderate Effect of Gender and Mindfulness Measurement Tools

There was a gender difference in the relationship between mindfulness and cognitive reappraisal; that is, the relationship between mindfulness and cognitive reappraisal was higher in men than in women, although the difference was statistically marginal. This is similar to previous research. In addition to gender differences in each of the two variables, there is evidence of gender differences in mindfulness and emotion regulation strategies [71,72]. For example, previous studies have found gender differences in the relationship between mindfulness and rumination, which is an ill-adaptive emotion regulation strategy. Specifically, the association was stronger in women than men [71]. This study found that a gender difference also applies to mindfulness and cognitive reappraisal. The difference is that the correlation between mindfulness and cognitive reappraisal, an adaptive emotion regulation strategy, in males is stronger than in females. However, whether this closer association can extend to mindfulness and other adaptive emotion regulation strategies are still needs more experimental support.

Due to the limited number of studies on the relationship between multiple dimensions of mindfulness and cognitive reappraisal and expressive suppression, the included studies failed to meet the requirements of meta-regression statistics. Thus, the moderating effect of the mindfulness dimension cannot be examined. Even though an insignificant correlation exists between mindfulness and expressive suppression, the study found that the use of mindfulness measurement tools can affect the relationship between mindfulness and expressive suppression. This indicates that different measurement methods of mindfulness will affect the results of the relationship between mindfulness and other variables. Therefore, it is necessary to comprehensively evaluate the existing measurement methods of mindfulness and improve them to obtain more consistent results. Different measurement tools of mindfulness may reflect different dimensions of mindfulness. Therefore, the moderating effect of mindfulness tools may be caused by the different dimensions of mindfulness measured.

### 4.2. Research Deficiencies and Prospects

Although the three-level meta-analysis technique was used in this study to systematically quantify the relationship between mindfulness and cognitive reappraisal and expressive suppression and draw some conclusions, there are still some things that could be improved. First, only two kinds of traits mindfulness and individual emotional regulation strategy orientation were discussed, without a comprehensive analysis of the relationship between mindfulness and other emotional regulation strategies. Secondly, the number of moderating variables included in the meta-analysis needed to be increased (some of them are less than 5) and did not meet the conditions of meta-regression analysis, resulting in some hypotheses cannot be directly verified and the low power of the results [65]. Third, experimental studies on the influence of experimentally controlled mindfulness and mindfulness training on emotion regulation strategies were not included in the analysis, making it impossible to conclude causality.

Future research should be carried out from the following aspects: First, strengthen the neuron physiological exploration of the relationship between mindfulness and other emotion regulation strategies, and comprehensively explore the relationship from the perspectives of brain imaging and event-related potential. Second, conduct a longitudinal study to explore the dynamic mechanism of mindfulness and other emotion-regulating strategies. Third, improve the relevant theories of mindfulness and emotion regulation, further clarify the difference and connection between the two from the connotation and denotation of the concept and then clarify their dimensional characteristics, and develop more ecological validity measurement tools.

## 5. Conclusions

The current three-level meta-analyses found a significant positive relationship between mindfulness and cognitive reappraisal and an insignificant relationship between mindfulness and expressive suppression of ordinary people. We also found that gender is an important moderator of the relationship between mindfulness and cognitive reappraisal; specifically, the relationship is closer for males than for females. Our finding clarified the magnitude of the relationship between mindfulness and the two emotion regulation strategies, and it also detected that gender is an important influence factor.

## Figures and Tables

**Figure 1 ijerph-20-01241-f001:**
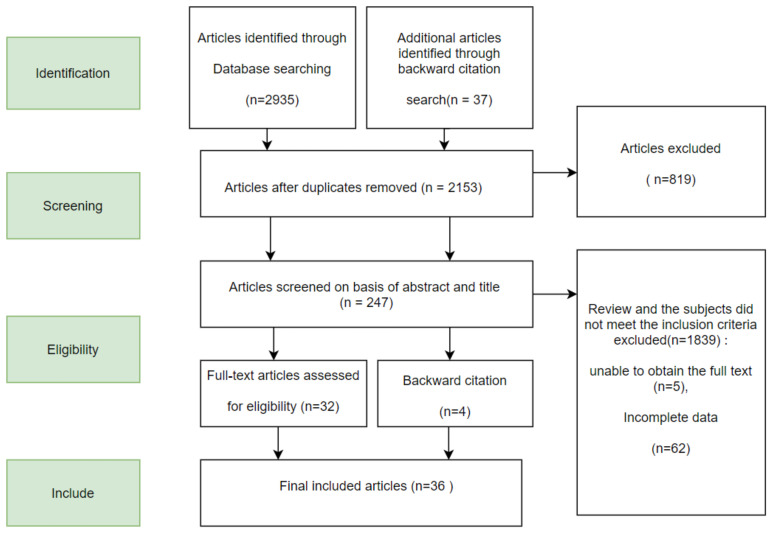
Flow chart of literature screening.

**Figure 2 ijerph-20-01241-f002:**
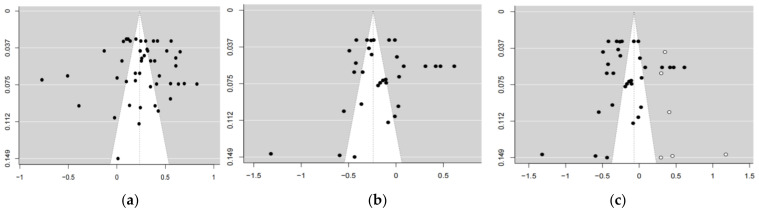
Funnel plots. (**a**) Funnel plot of mindfulness and cognitive reappraisal; (**b**) funnel plot of mindfulness and expressive suppression; (**c**) funnel plot of mindfulness and expressive suppression after trim-fill corrected. Note: The vertical coordinate represents the standard error, the horizontal coordinate represents Fisher’s *z* value transformed by *r*, and the hollow dot in the right-most figure represents the effect size that needs to be supplemented after the trim-fill method is adopted.

**Table 1 ijerph-20-01241-t001:** The combined effect size of mindfulness and cognitive reappraisal and expressive suppression.

Analysis Methods	K	n	*Q* (*df*)	*P*	*r*	95% CI	Level 2 *I*^2^	Level 3 *I*^2^	Total *I*^2^
A	32	50	898.87 (31)	<0.001	0.23	[0.13, 0.33]	28.2%	63.05%	91.25%
B		50	1460.65 (49)	<0.001	0.23	[0.14, 0.31]			97.98%
C	23	32	898.87 (31)	<0.001	−0.24	[−0.36, −0.11]	26.2%	71.19%	97.39%
D		38	1227.38 (37)	<0.001	−0.07	[−0.20, 0.06]			98.34%
E		32			−0.02	[−0.19, 0.04]			

Note: A = three-level meta-analysis of mindfulness and cognitive reappraisal; B = classical meta-analysis of mindfulness and cognitive reappraisal; C = three-level meta-analysis of mindfulness and expressive suppression; D = trim-fill corrected meta-analysis of mindfulness and expressive suppression; E = robust Bayesian meta-analysis model of mindfulness and expressive suppression. K = number of studies, n = number of effect sizes.

**Table 2 ijerph-20-01241-t002:** Comparison between the three-level random effects model and the two-level random effects model.

Models	Mindfulness and Cognitive Reappraisal				Mindfulness and Expressive Suppression			
	AIC	BIC	LRT	*p*	AIC	BIC	LRT	*p*
Level (1,2,3)	35.33	41.01			15.54	19.85		
Level (1,2)	33.25	37.04	0	1	27.37	30.24	13.83	<0.001
Level (1,3)	371.87	375.65	338.54	<0.001	150.03	152.91	136.48	<0.001

Note: AIC = Akaike information criterion; BIC = Bayesian information criterion; LRT = likelihood ratio test.

**Table 3 ijerph-20-01241-t003:** Meta-regression analysis.

Meta-Regression Models	K	n	*Q* _E_ *(df)*	*p*	*b*	95% CI	*F*	*p*
Mindfulness and cognitive reappraisal								
age	25	37	1135.85 (35)	<0.001	0.06	[−0.08, 0,19]	*F* (1,35) = 0.79	0.38
culture (Eastern culture as reference)	32	50	1357.82 (48)	<0.001			*F* (1,48) = 2.30	0.14
Western culture	20	32			0.17	[−0.05, 0.39]		0.14
gender (women divided by the total number)	29	44	1335.28 (42)	<0.001	−0.02	[−0.03, 0.00]	*F* (1,42) = 3.08	0.09
mindfulness measurement tools (FFMQ as reference)	27	44	1316.74 (42)	<0.001			*F* (1,42) = 2.39	0.13
MAAS	14	19			−0.16	[−0.36, 0.05]		0.13
emotion regulation measurement tools (CERQ as reference)	26	42	996.88 (40)	<0.001			*F* (1,40) = 2.26	0.14
ERQ	19	29			0.23	[−0.08, 0.53]		0.14
study quality (weak as reference)	32	50	1378.05 (47)	<0.001			*F* (2,47) = 1.73	0.19
middle	16	22			−0.16	[−0.49, 0.18]		0.36
strong	4	6			0.07	[−0.33, 0.48]		0.71
Mindfulness and expressive suppression								
age	18	27	780.80 (25)	<0.001	0.01	[−0.16, 0.14]	*F* (1,25) = 0.02	0.89
culture (Eastern culture as reference)	23	32	874.78 (30)	<0.001	−0.10	[−0.40, 0.20]	*F* (1,30) = 0.48	0.49
Western culture					−0.10	[−0.40, 0.20]		0.49
gender (women divided by the total number)	20	29	812.20 (27)	<0.001	−0.06	[−0.18, 0.07]	*F* (1,27) = 0.86	0.37
mindfulness measurement tools (FFMQ as reference)	18	26	728.19 (24)	<0.001			*F* (1,24) = 4.35	0.05
MAAS	13	14			−0.35	[−0.70, −0.01]		0.05
Mindfulness dimension (no-judgment dimension as reference)	23	24	521.24 (22)	<0.001			*F* (1,22) = 1.22	0.28
single dimension	19	20			−0.22	[−0.62, 0.20]		0.28
study quality (weak as reference)	23	32					*F* (2,29) = 0.42	0.66
middle	15	21			0.04	[−0.16, 0.23]		0.69
strong	4	6			−0.10	[−0.41, 0.20]		0.49

Note: K = number of studies, n = number of effect sizes; *b* = coefficient of meta-regression; 95%CI = 95% confidence intervals of *b. F = F* test value of the regression model.

## Data Availability

The main information on studies included in our meta-analysis was provided in Appendix A. All correspondence and requests for full data should be addressed to S.Z.

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
