# Peer review of "Linking Cognitive Reappraisal and Expressive Suppression to Mindfulness: A Three-Level Meta-Analysis"

_ijerph, 2023, doi:10.3390/ijerph20021241_

Round 1
Reviewer 1 Report
This is an interesting meta-analysis that synthesizes correlation statistics from 36 studies with effect sizes of 83. The article is well written and addresses a major gap by linking cognitive reappraisal and expressive suppression to mindfulness. Although the study has several merits, there are very few issues the authors must address before the work can be considered fit for publication.
Topic: The topic will be clearer if written as "Linking cognitive reappraisal and expressive suppression to mindfulness: A three-Level meta-analysis"
Authors generally need to proofread their work for grammatical concerns. Authors should also check the attached document for specific comments marked in the article.

Author Response
This is an interesting meta-analysis that synthesizes correlation statistics from 36 studies with effect sizes of 83. The article is well written and addresses a major gap by linking cognitive reappraisal and expressive suppression to mindfulness. Although the study has several merits, there are very few issues the authors must address before the work can be considered fit for publication.
Topic: The topic will be clearer if written as "Linking cognitive reappraisal and expressive suppression to mindfulness: A three-Level meta-analysis"
Response: thank you for your advice and we have adjusted the title to"Linking cognitive reappraisal and expressive suppression to mindfulness: A three-Level meta-analysis"
Authors generally need to proofread their work for grammatical concerns. Authors should also check the attached document for specific comments marked in the article.
Response: All the authors did the proofreading work for grammatical and spelling concerns, and the errors have been corrected. The attached document for specific comments marked in the article has also been checked.

Reviewer 2 Report
The authors provided a very detailed and methodologically high qualitative meta-analysis regarding the association between mindfulness and two emotion regulation strategies, i.e., cognitive reappraisal and expressive suppression as well as key influential variables. They found that mindfulness is associated with cognitive reappraisal (small to medium effect size) but not with expressive suppression (based on a trim-filled estimate). The meta-analysis seems to be conducted thoroughly and with expertise. I have only minor issues to point out, mainly the addition of further information to replicate the findings. Additionally, the quality of underlying studies as well as the certainty of evidence (like the GRADE approach in Cochrane analyses) was not examined which must be discussed further and may question the here presented results.
Section 2.1: I am not quite sure that I could replicate your article search with the provided search terms. Were other field functions use like MESH terms or others? I would recommend that you stick to the PRISMA guidelines to provide a thorough search strategy.
Section 2.1: Who conducted the abstract and full-text screening? Were these screenings independently by multiple screeners? I guess the methods section needs some additional information to potentially replicate or even reproduce the study results.
ll. 274 – 279: I am a bit confused, the numbers don’t seem to add up correctly. Can you help me clarify this issue? 2936 (which, by the way, is different from the number in Figure 1) – 783 duplicates – 1906 studies that did not fit – 207 articles excluded after full-text screening = 40 articles, not 36. Are there some typos?
ll. 280 – 284: I am not quite sure what you mean by that.
ll. 353 – 355: The decision whether to use a random or fixed effect meta-analysis must be made a priori based on the fact if the sampled effect sizes are likely from the same population of different populations. In the latter case, random effects meta-analyses are required. This decision process should be made transparent in the methods section and not the results section.
ll. 370 – 373: The notion that your hypothesis 1 is supported by the results relies solely on the fact that you respect the trim-fill corrected meta-analysis of mindfulness and expressive suppression. I wonder if that is a particularly good and stable method for this kind of application. I know that it is widely used, but you could add more information if this method yields stable or confident results (also see https://doi.org/10.31234/osf.io/75bqn)
Section 3 and 4: The quality of studies as well as the quality of evidence was not determined. Hence, the results of this meta-analyses may not be reliable (if, for instance, the underlying study quality is very low)
Author Response
the authors provided a very detailed and methodologically high qualitative meta-analysis regarding the association between mindfulness and two emotion regulation strategies, i.e., cognitive reappraisal and expressive suppression as well as key influential variables. They found that mindfulness is associated with cognitive reappraisal (small to medium effect size) but not with expressive suppression (based on a trim-filled estimate). The meta-analysis seems to be conducted thoroughly and with expertise. I have only minor issues to point out, mainly the addition of further information to replicate the findings. Additionally, the quality of underlying studies as well as the certainty of evidence (like the GRADE approach in Cochrane analyses) was not examined which must be discussed further and may question the here presented results.
Response: thank you for your valuable advice. We added the quality rating to our paper, and also detected if the quality influenced the pooled effect size. Please see lines 302-313, and table 3.
Section 2.1: I am not quite sure that I could replicate your article search with the provided search terms. Were other field functions use like MESH terms or others? I would recommend that you stick to the PRISMA guidelines to provide a thorough search strategy.
Response: We specified the search strategy and the terms used in databases: “Search for foreign papers in Web of Science, PubMed, and Elsevier Science Direct database for keywords. Two authors (Zhou and Xu) extracted the high-frequency keywords in the databases, and other keywords were evaluated independently by two authors (Zhou and Xu). All discrepancies were resolved via discussion. The search string included the following elements: ‘mindfulness, cognitive reappraisal, and expressive suppression’. Synonymous terms such as ‘mindful*, meditate*, emotion regulation, emotion adjust, express*, suppress*’ were combined with the Boolean ‘OR’. These three concepts were then combined with the Boolean ‘AND’.“. Please see lines 252-260.
Section 2.1: Who conducted the abstract and full-text screening? Were these screenings independently by multiple screeners? I guess the methods section needs some additional information to potentially replicate or even reproduce the study results.
Response: Two authors (Zhou and Xu) independently conducted the abstract and full-text screening, and any disagreements on the criterion were resolved by consensus. We added this information in lines 288-290.
274 – 279: I am a bit confused, the numbers don’t seem to add up correctly. Can you help me clarify this issue? 2936 (which, by the way, is different from the number in Figure1) – 783 duplicates – 1906 studies that did not fit – 207 articles excluded after full-text screening = 40 articles, not 36. Are there some typos?Response: I am sorry for this mistake, and we have corrected it in the paper.
280 – 284: I am not quite sure what you mean by that.
Response: sorry for confusing you, we have deleted these ambiguous sentences.
353 – 355: The decision whether to use a random or fixed effect meta-analysis must be made a priori based on the fact if the sampled effect sizes are likely from the same population of different populations. In the latter case, random effects meta-analyses are required. This decision process should be made transparent in the methods section and not the results section.
Response: We added the reason for use of the random model in line 370-372: “Given that the study samples included in our meta-analysis are from different cultures, ages, and different backgrounds, also other potential moderators may induce heterogeneity, it is reasonable to conduct a random effects model.” also see line 382-383.
370 – 373: The notion that your hypothesis 1 is supported by the results relies solely on the fact that you respect the trim-fill corrected meta-analysis of mindfulness and expressive suppression. I wonder if that is a particularly good and stable method for this kind of application. I know that it is widely used, but you could add more information if this method yields stable or confident results (also see https://doi.org/10.31234/osf.io/75bqn)
Response: Thank you for your advice. We read the article
(https://doi.org/10.31234/osf.io/75bqn) carefully and adopted the Robust Bayesian Meta-analysis method introduced by that article to verify the robustness of the result got by the trim-fill method and offer a more precise pooled effect size. Please see line356-362, 391-392, and table 2.
Section 3 and 4: The quality of studies as well as the quality of evidence was not
determined. Hence, the results of this meta-analyses may not be reliable (if, for instance, the underlying study quality is very low)
Response: We added the quality rating statement to our paper, and also detected if the quality influenced the pooled effect size. The meta-regression suggested that article quality have an insignificant influence on pooled effect size. Please see lines 302-313, and table 3.
